# Macromolecule conformational shaping for extreme mechanical programming of polymorphic hydrogel fibers

Xiao-Qiao Wang [1], Kwok Hoe Chan[1], Wanheng Lu[1], Tianpeng Ding[1], Serene Wen Ling Ng [1], Yin Cheng[1], Tongtao Li[1], Minghui Hong[1], Benjamin C. K. Tee [1,2] & Ghim Wei Ho [1,2 ✉]

Mechanical properties of hydrogels are crucial to emerging devices and machines for wearables, robotics and energy harvesters. Various polymer network architectures and interactions have been explored for achieving specific mechanical characteristics, however, extreme mechanical property tuning of single-composition hydrogel material and deployment in integrated devices remain challenging. Here, we introduce a macromolecule conformational shaping strategy that enables mechanical programming of polymorphic hydrogel fiber based devices. Conformation of the single-composition polyelectrolyte macromolecule is controlled to evolve from coiling to extending states via a pH-dependent antisolvent phase separation process. The resulting structured hydrogel microfibers reveal extreme mechanical integrity, including modulus spanning four orders of magnitude, brittleness to ultrastretchability, and plasticity to anelasticity and elasticity. Our approach yields hydrogel microfibers of varied macromolecule conformations that can be built-in layered formats, enabling the translation of extraordinary, realistic hydrogel electronic applications, i.e., large strain (1000%) and ultrafast responsive (~30 ms) fiber sensors in a robotic bird, large deformations (6000%) and antifreezing helical electronic conductors, and large strain (700%) capable Janus springs energy harvesters in wearables.

[1] Department of Electrical and Computer Engineering, National University of Singapore, 4 Engineering Drive 3, Singapore 117583, Singapore. [2] Department of Materials Science and Engineering, National University of Singapore, 9 Engineering Drive 1, Singapore 117575, Singapore. ✉email: elehgw@nus.edu.sg

Hydrogels are rapidly gaining traction for use in wearable/on-skin devices and machines, such as sensors[1,2], actuators[3–5] and energy harvesters[6–8]. Owing to the excellent softness, biocompatibility, stimulus diversity, and good compatibility with electrical and ionic components[9–14], hydrogels are the ideal matrix for bio-integrated smart systems. Hydrogels that possess specific mechanical properties is one of the central prerequisites for diverse hydrogel applications[15,16]. However, thus far, the mechanical programming[17] of hydrogel and its realistic implementation in devices remain underexplored. The difficulties mainly arise from the three aspects. For one thing, the conventional polymer networks of hydrogels are composed of randomly cross-linked polymer chains with covalent bonds, in which entanglements and physical cross-links of the polymer chains are negligible. The formed hydrogels are usually fragile, and the integrated devices suffer low mechanical robustness and low stretchability. For another, the unconventional polymer networks of hydrogels in terms of network architectures and interactions among polymer chains have been widely explored in the past two decades[18]. Representative examples include interpenetrating polymer networks based on covalent and physical hybrid cross-linking strategies[19–21], polymer networks with slide-ring cross-linkers[22], and polymer networks with high-functionality cross-linkers, such as crystalline domains[23,24], hydrogen bonds[25], ionic bonds[26] and nano-/microparticles[27–30]. Remarkable mechanical properties, such as high toughness, strength, resilience and interfacial toughness have been achieved. However, these sophisticated polymer networks usually need meticulous design of the composition components, sequential or prolonged polymerization, or cumbersome pre/post-polymerization treatment. The fabrications mostly depend on custom-designed molds, which are of low throughput, limited in shape control and difficult in upscaling. So far, few material strategies have been reported that allow printing[31,32] or spinning[33,34] of mechanically robust hydrogels. Lastly, current hydrogel devices with stimulus-responsiveness can durably work only in aqueous environments. As such, encapsulation using elastomers is employed to alleviate the dehydration of hydrogels in air, but introduce interfacial assembly and adhesion issues that existed in the heterogeneous material systems[15,35,36]. To advance mechanical properties of hydrogels toward practical translation into functional devices calls for new polymer material approaches as well as advanced fabrication strategies.

Here, we introduce a simple, scalable approach for mechanical programming of polymorphic hydrogel fibers, enabling extreme mechanical tunability of single-composition hydrogel fibers and their utilization for extraordinary electronic devices. Our method shows that the conformation of a poly(acrylic acid) based polyelectrolyte macromolecule can be synthetically engineered from compactly coiled to extended, aligned states based on a pH-dependent antisolvent phase separation process (left of Fig. 1a). The pH determines the original conformations of the polyelectrolyte macromolecules, while the phase separation drives aggregation of the macromolecules and generates densely entangled macromolecule networks. The resulting hydrogel microfibers represent a new type of unconventional polymer network based on the single-composition entangled polyelectrolyte macromolecule with controllable conformations. And tunable mechanical properties ranging from softness to ultrahigh rigidity, brittleness to ultrastretchability, and plasticity to anelasticity and elasticity, can be programmed (Fig. 1b, c). Moreover, the heterogeneous mechanical properties resulting from the complex macromolecule architectures can be arbitrarily layered in one-step via our approach, thereby generating hydrogel fiber devices with readily customizable shapes. Polymorphic hydrogel fibers of fibers/ribbons, Janus fibers, multilayered fibers, core-shell fibers, helical Janus fibers, Janus springs and beyond, can be fabricated (right of Fig. 1a). We demonstrate the programing and manufacturing of ultra-large strain capable hydrogel fiber devices with functionalities spanning across sensors, conductors and generators. The hydrogel devices operate durably in ambient air with extended shelf life and intriguingly, this remains the case even at sub-zero temperature and under extreme deformation state.

## Results

**Macromolecule conformation shaping of hydrogel microfibers.** We exploit conformational tuning[37] of the polyelectrolyte macromolecule based on a pH-dependent antisolvent phase separation-induced hydrogel filamentation process. The poly(acrylic acid) based polyelectrolyte, commercially available sodium polyacrylate (PANa) of molecular weight ($\sim 3 \times 10^7$ Da) was chosen as the starting material. Especially, the high molecular weight of PANa is required for polymer networks with entangled long macromolecule chains, which are promising for acquiring wide-range tunable mechanical properties when the coiling state of the macromolecule chains changes. In addition, PANa polyelectrolyte with abundant carboxylate groups is hygroscopic, capable of stable water retention in the ambient air (65% humidity). The viscosity of 3.5 wt% polyelectrolyte hydrogel dopes at different pH of 3.95 to 13.97, correspond to the strain dependent rheological results. Phase separation occurs at pH 3.10 (Supplementary Fig. 1a). The transparent dopes exhibit viscoelastic behavior, and the viscosity decreases with increasing applied shear rate (Supplementary Fig. 1b), which is desirable for the smooth extrusion of spinning dopes. As the pH increases, conformation changes from coiled to extended states in the dopes of the same concentration are induced, accompanied by the affinity changes of the macromolecules to water molecules. This leads to a pH-dependent viscosity and moduli (see details in Supplementary Figs. 2 and 3). During the filamentation process, methanol was used as the antisolvent to drive the aggregation and conformation change of polyelectrolyte macromolecules. Upon extrusion into methanol bath, the polyelectrolyte macromolecules undergo fast dehydration as water molecules in the network are captured by methanol due to the higher affinity of methanol to water than that of carboxylate groups (left of Fig. 1a). Supplementary Movie 1 presents the continuous process of hydrogel filamentation and collection of microfibers. During the dehydration and phase separation, the macromolecules of different pH aggregate and experience specific conformation changes depending on their own water affinity. The dehydration process of those pH-varied hydrogel dopes takes 150 to 250 s, and all of the water in the dopes diffuses into the methanol bath and the microfibers become rigid upon completion of phase separation. Dopes of the lowest pH 3.95 can be solidified fastest (Supplementary Movie 2), due to the weakest bonding of protonated polyelectrolyte macromolecules to water molecules. Meanwhile, the phase separation-induced aggregation of the macromolecule chains generates densely entangled macromolecule networks, thereby engendering materials toughness. After that, the solidified microfibers are exposed to the ambient environment, and the porous macromolecule networks undergo spontaneous structural collapse and self-assembly. Cross-sectional areas of the resultant hydrogel fibers are in the range of $\sim 0.08$ to $0.2\ mm^2$. Importantly, those microfibers regain water from the ambient air due to the hygroscopic nature of PANa, and their equilibrated water contents depend on conformations of the polyelectrolyte macromolecules, which vary from $\sim 12$ to 40 wt% with increasing pH (Supplementary Figs. 4 and 5).

Through this process of hydrogel filamentation, single-composition entangled macromolecule networks of controlled macromolecule conformations can be prepared, thereby

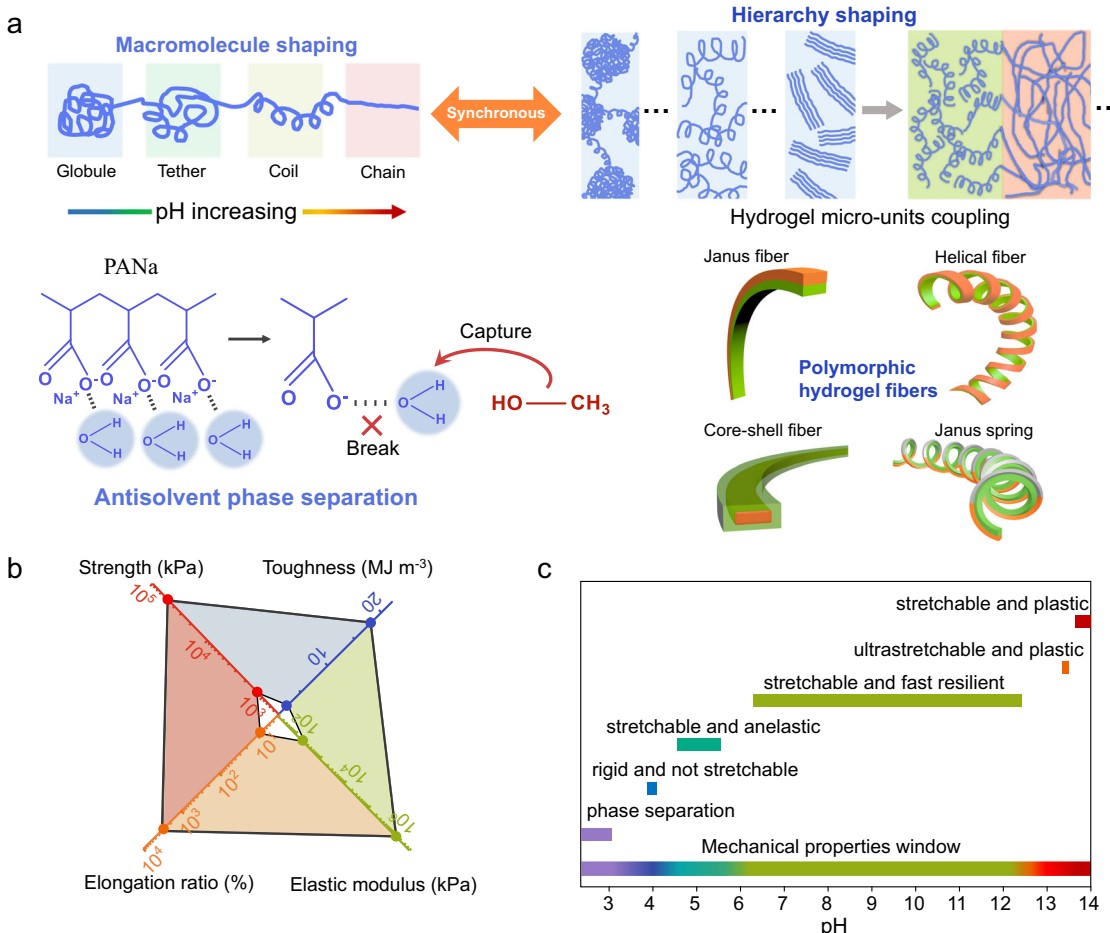

**Fig. 1 Extreme mechanical property tuning and shaping of polymorphic hydrogel fibers. a** Conformational shaping of the polyelectrolyte macromolecule based on the pH-dependent antisolvent phase separation to generate sodium polyacrylate (PANa) hydrogel microfibers of complex network architectures (left), and their microfiber units coupling to produce polymorphic hydrogel fibers (right). **b** Mechanical parameters scope of the hydrogel microfibers with pH from 3.95 to 13.97. **c** Mechanical gradation of the microfibers with increased pH.

continuously modulating various mechanical properties of the hydrogel microfibers, including modulus, strength, stretchability and resilience properties (Fig. 1b, c). The macromolecule conformations evolve from tightly coiled to extended, aligned states with increasing pH (Supplementary Fig. 6), corresponding to the various nanostructured hydrogel networks shown in atomic force microscopy (AFM) images. Those include aggregated globules, aggregated coils, extended interpenetrated coils, partially aligned chains, and compactly aligned chains (Fig. 2a and Supplementary Fig. 7). Confocal and laser-scanning microscope images present the microscale hydrogel networks and 3D surface microstructures (Supplementary Figs. 8 and 9). The macrostructures of the fibers were studied by optical microscope in polarization and transmission modes (Fig. 2b and Supplementary Fig. 10). The brilliant interference sequent color changes from blue, yellow, pink to green in the polarized optical microscopy images fibers of pH ranging from 13 to 14 (Supplementary Fig. 10), match the order of colors in the Michel-Levy color chart as the birefringence increase[38], confirming the gradual macromolecule alignment. Notably, the obscured interference color of pH 9.14 fiber transformed into brilliant color when it was stretched. The sequent color changes from 100 to 800% strain are attributed to continuous increasing of macromolecule orientation along the fiber direction induced by stretching (Supplementary Fig. 11), and those colors would instantly disappear upon strain release. The infrared spectra of the dried hydrogel fibers (Supplementary Fig. 12) show diminishing C=O stretching band of –COOH at 1700 cm$^{-1}$ and enhancement of asymmetric and symmetric –COO$^{-}$ stretches at respective 1546 and 1395 cm$^{-1}$ as pH increases, suggesting different protonation states of the conformation varied macromolecules. Small-angle X-ray scattering (SAXS) results reveal the evolution of delicate structures below 100 nm. Uniform, continuous polymer networks of low scattering contrast in the moderate pH range display weak scattering signal whereas the networks containing aggregated domains exhibit strong scattering peaks (Supplementary Fig. 13a, c). The stretched SAXS pattern perpendicular to the fiber axis of pH 13.97 microfiber signifies the presence of aligned nanostructures, as proven by the AFM results. The extension and reorientation of macromolecule chains in the pH 9.14 microfiber under stretching produce scattering centers perpendicular to the stretching direction[39], and the continuously increased macromolecule orientation within 800% tensile strain leads to intensified and narrowed scattering in the perpendicular direction (Supplementary Fig. 13b, d–f). We used Wide-angle X-ray scattering measurements to determine the macromolecule orientation parameters at different strains (Supplementary Fig. 14), and the calculated orientation factor ($f$) is 0.42 at 0% strain, which increases to 0.67 at 100% strain, 0.80 at 400% strain, and 0.91 at 800% strain. The increase in $f$ indicates that the orientation of macromolecule chains along the microfiber axis continuously improves with an increasing stretching ratio.

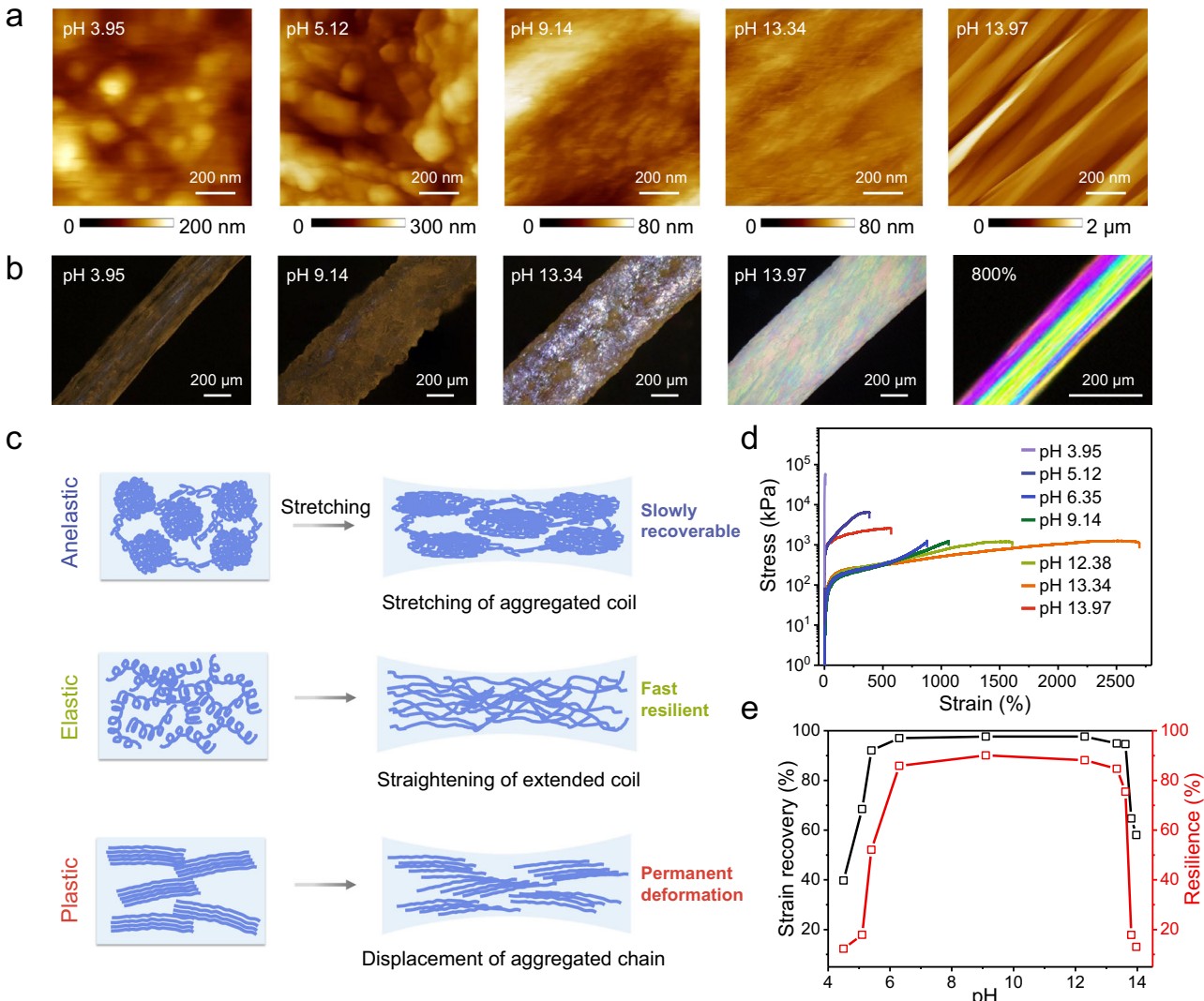

**Fig. 2 Structural and mechanical characterization for the single-composition polyelectrolyte hydrogel microfibers of evolving macromolecule conformations. a** AFM images of hydrogel microfibers of different pH. **b** POM images of microfibers of different pH, and pH 9.14 microfiber at 800% strain. **c** Schematic illustration of the response of different macromolecule network architectures to the mechanical loading. **d** Strain-stress curves of the hydrogel microfibers. **e** Mechanical resilience and strain recovery of the microfibers under a 200% strain cycle.

Diverse macromolecule conformational shaping realize continuous and wide-range mechanical programming of the single-composition hydrogel microfibers, in terms of both stretchability and resilience properties. The successive molecular tuning comprises of (i) aggregated macromolecule coils with strong intermolecular interactions of low stretchability but high strength, and the resulting fiber is anelastic with slow tensile strain and strength recovery; (ii) extended, interpenetrated coils that are highly stretchable and elastic with reversible random entanglement to orientated state, and deformation of the fiber can be instantly recovered; (iii) aggregated, aligned chains network of low stretchability and plastic deformation due to the tight bonding of extended macromolecule chains (Fig. 2c).

As measured, hydrogel microfiber of the lowest pH is stiff and not stretchable and the strength continues to decrease alongside stretchability enhancement until the pH increases to 13.34 (Fig. 2d and Supplementary Fig. 15), when extended macromolecules start to align. The pH 13.34 microfiber is soft and ultrastretchable with a breaking strain reaching 2693%, yielding the highest toughness of 20.3 MJ m$^{-3}$. As the polymer chains further align and aggregate with increased pH, the strength

increases while stretchability decreases. The pH-dependent mechanical parameter (Supplementary Fig. 16 and Supplementary Table 1), correspond to the scope of elongation ratio of 105% ± 2% to 2630% ± 120%, tensile strength of 1210 ± 120 kPa to 47 ± 5 MPa, modulus of 240 ± 30 kPa to 2050 ± 370 MPa, and toughness of 1.7 ± 1.1 MJ m$^{-3}$ to 17.8 ± 1.6 MJ m$^{-3}$. Besides, strain rate and humidity effect were studied (Supplementary Figs. 17 and 18). Polyelectrolyte networks of the hydrogel fiber keep stable water content for steady mechanical performance even after being stored in the ambient environment for 5 months (Supplementary Fig. 18b).

Mechanical resilience properties were studied via loading-unloading tests (Supplementary Fig. 19). Hydrogel microfibers in the moderate pH range (6.35–12.38) are fast resilient, with strain recovery and resilience of pH 9.14 fibers reaching 97.7% and 90.1%, respectively (Fig. 2e). Tensile tests of 1000 cycles at different strains show very small residual strain even with 800% cycled strains, and the strength regresses in the first 100 cycles but degrades marginally in the following 800 cycles (Supplementary Figs. 20 and 21). The fabricated hydrogel ribbon (Supplementary Movie 3) exhibits similar mechanical characteristics to that of

microfibers form (Supplementary Fig. 22). Large hysteresis and residual strain were observed for microfibers of both low and high pH regimes. However, unlike plastic deformation experienced by the fibers of pH above 13, the residual tensile strain and strength of the fibers at around pH 5 can be automatically recovered after 1–2 h, suggesting their anelastic properties (Supplementary Fig. 23). Supplementary Movie 4 presents the three different mechanical resilience properties. The summary chart in Fig. 1c delineates the mechanical gradation of the hydrogel microfibers in a full pH range.

**Processability of macromolecule conformation tunable hydrogel fibers**. Coupling of component units with heterogeneous properties, such as mechanical features[40], electronic/magnetic properties[41,42], or responsiveness to external fields[43], could evolve complex outputs. Here, the polyelectrolyte macromolecules of varied conformations offer the single-composition hydrogels various mechanical properties, and meanwhile, the hydrogel matrix could encompass a variety of active fillers to enable functional heterogeneity. Monolithic, layered hydrogel devices can be prepared in one-step, which consist of conformations varied macromolecules based hydrogel microfiber units with conformal and anti-fracture interfaces. Such a material approach bypasses the challenge of forming strong chemical interfacial bonding between hydrogel and antagonistic materials in the fabrication of conventional hydrogel devices, and enables mechanical programming of functional hydrogel devices with adaptive form factors. (Fig. 3a). We show the programming of mechanical resilience mismatched Janus microfibers for diameter-controllable spring-like structures, adaptive for ultrastretchable hydrogel electronics. A parallel-axial dual-spinneret system is devised to layer two microfibers in a stacked structure, forming Janus hydrogel fibers of arbitrary length. The fast resilient pH 12.38 hydrogel phase serves as the elastic substrate while the counterpart ultrastretchable, plastic pH 13.34 hydrogel phase is doped with single-walled carbon nanotubes (SWCNTs) to evolve heterogeneity in electrical conductivity and mechanical resilience (Fig. 3b). Here, SWCNTs are utilized as functional fillers to produce high electrical conductivity and thermoelectric (TE) properties. Scanning electron microscope images show that SWCNTs are well dispersed within the hydrogel networks (Supplementary Fig. 24). The pH 13.34 composite hydrogel with 20 wt% SWCNTs has a conductivity of 88.7 S m$^{-1}$, capable of tunable permanent plastic deformation within 900% prestrain (Supplementary Figs. 25 and 26). Upon release from stretching, the elastic pH 12.38 microfiber unit in the Janus fiber contracts to its original length while plastic deformation is produced in the pH 13.34 microfiber unit, and the resulting stress induces helical structure formation. Supplementary Movie 5 presents the continuous fabrication of Janus hydrogel fibers, and conformally bonded interface forms through the synchronous phase separation-induced filamentation of the two phases and the spontaneous entanglement of macromolecules at the interface (Fig. 3c). Supplementary Fig. 27 and Movie 6 show the formation process of well-bonded interfaces at different flow rates. Diameter-controllable hydrogel springs can be easily programmed by applying 200–900% prestrain to the Janus fibers (Fig. 3d and Supplementary Movie 7). The calculated resilience of two microfiber units subjected to a 900% strain cycle are 88.3% and 20.1%, respectively (Supplementary Fig. 28). Supplementary Fig. 29 depicts the dimensional change of the Janus hydrogel fiber as increasing prestrain produces helical fiber of a smaller diameter and a greater number of turns per centimeter. The diameters of the helices are in the range of 1.8 to 6.7 mm (Fig. 3e and Supplementary Fig. 30). Besides, core-shell hydrogel fibers were

fabricated by utilizing a coaxial spinneret (Supplementary Fig. 31), and the multi-channel parallel-axial spinneret could be assembled to fabricate multilayered hydrogel fibers. It is worth mentioning that our material approach for mechanical programming and manufacturing of hydrogel devices is not only compatible with mature hydrogel wet-spinning technology to construct elaborate fibers, but also extendable to the contemporary direct ink writing to produce patterned matrices (Supplementary Fig. 32).

**Large strain and ultrafast responsive ionic hydrogel fiber strain sensor**. Mechanical programming of the macromolecule conformation tunable hydrogels enables a new class of all-soft, ultra-large strain capable fiber microelectronic devices with customizable functions ranging from sensing, conducting to energy harvesting. So far, hardly any known conductive hydrogel fibers can stably and repeatedly sense high-speed motions in a broad working range. Here, highly stretchable, resilient pH 12.38 ionic hydrogel fibers are capable of reliably and durably monitoring large-strain motions (up to 1000%) from slow-adaptive mode to fast repeated motions in a high-frequency dynamic mode. Supplementary Fig. 33a depicts the relative change of resistance ($\Delta R/R_0$) with the applied strain ($\varepsilon$) from 50 to 1000% in a slow motion mode. The resistance change is constant without obvious creep under a fixed deformation (Supplementary Fig. 33b). The relationship between the resistance and strain is a quadratic equation, which is fitted with two linear equations. The gauge factor (GF = ($\Delta R/R_0$)/$\varepsilon$) is 2.29 at 0–400% strain and 4.41 at 400–1000% strain (Fig. 4a). Furthermore, high-speed mechanical response of the strain sensor was measured by transforming circular motions of the motor into high-frequency reciprocating rectilinear motions (Supplementary Fig. 34a and Movie 8). The maximum monitoring frequency of 100% strain reaches 8.6 Hz, the resistance change exhibits a slight drop as the stretching frequency increases from 2.7 to 8.6 Hz (Supplementary Fig. 34b). The frequency dependence resistance change at different strains were measured. The sensing frequency can reach up to 5.7 Hz as measured, even at 600% strain (Fig. 4b and Supplementary Fig. 34c). Moreover, the hydrogel fibers exhibit a stable change in resistance under thousands of large strain cycles (Fig. 4c and Supplementary Fig. 35), suggesting the fast mechanical resilience and stable electronic performance in the ambient environment. We demonstrate the application of the hydrogel fiber as a lightweight sensor for wireless monitoring of wing flapping motions of a robotic bird. The fiber was installed between its head and one of the wings at the upmost position in the released state; when the wing was pushed up and down, the fiber was circularly stretched and released, producing signals induced by resistance change that were wirelessly transmitted to the user interface via Bluetooth (Fig. 4d). The robotic bird with a fully integrated fiber sensor, bluetooth chip and power source, is able to wirelessly and remotely deliver its motion dynamics (Fig. 4e). Figure 4f presents the wireless and stable signals that show motion states of the bird, including static wing at different positions, and high-speed wing flapping at different frequencies. Signals of ~16.5 Hz (response/recovery time ~30 ms) are recorded when the robotic bird takes off and flies (Supplementary Fig. 36 and Movie 9).

**Extremely large deformations and low temperatures tolerant helical hydrogel electronic conductor**. Helical hydrogel fibers with controllable helix can be easily generated by applying different prestrains along the length direction of the Janus fiber. The hydrogel springs are especially apt for ultrastretchable, all-soft electrodes/conductors. Figure 5a shows the various helical structures with the same and different diameter coils. The stretching limit of the helical conductors increases from ~3800 to 6000% as the diameter increases from 1.8 to 6.7 mm, along with

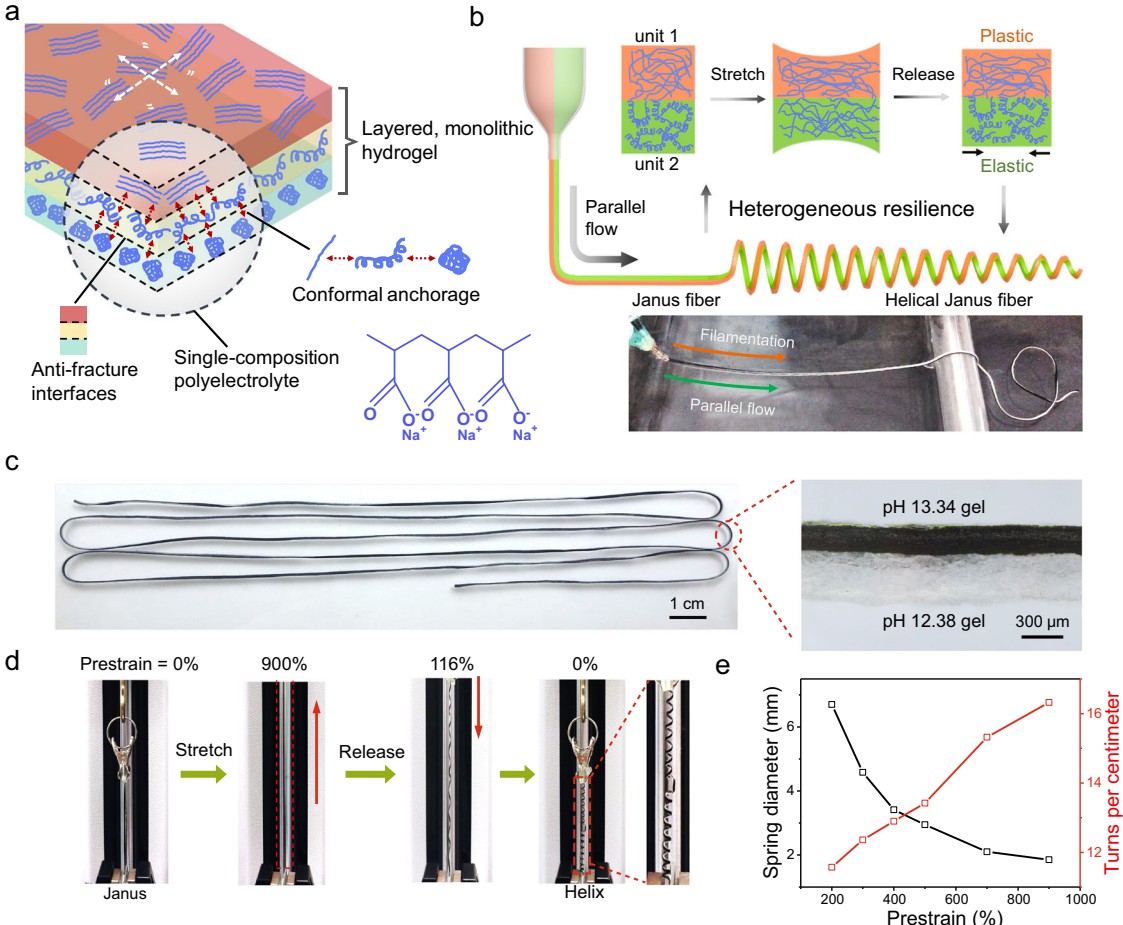

**Fig. 3 Processability of the macromolecule conformation tunable hydrogels for monolithic, polymorphic hydrogel fiber devices. a** Schematic illustration of layered, monolithic hydrogels of varied macromolecule conformations. **b** Schematic illustration for the preparation of helical Janus hydrogel fiber (top), and fabrication of continuous Janus hydrogel fiber via a parallel-axial dual-spinneret (bottom). **c** Photograph of the Janus hydrogel fiber of >1 meter in length and optical microscope image showing the conformal interface of layered hydrogel microfibers. **d** Formation of a helical Janus fiber in response to 900% prestrain. **e** Spring diameter and number of coils of helical Janus fibers versus applied prestrain.

the increase of the spring index from 4.1 to 11.5 (Fig. 5b). The helical conductors with diameters of 2.1 and 2.9 mm show negligible residual strains at respective 4000 and 5000% strain cycling (Fig. 5c). Figure 5d displays relative resistance changes of the conductors in response to tensile strain. For the helical conductor with a diameter of 2.9 mm, the resistance slowly increases by 8.3% when stretched to 1200% strain, while resistance rapidly increases after straightened. In the process of being straightened, the helical conductor first experiences uncoiling (0–310%) accompanied by a relatively fast resistance increase, and then coil straightening (310–1000%) with a slower resistance increase (Fig. 5e). Cyclic 1000% strain test proves high durability of the helical conductor (Supplementary Fig. 37).

We also investigated the antifreezing property of the helical conductor at sub-zero temperatures. Differential scanning calorimetric measurements of the pH 12.38 ionic hydrogel fiber and pH 13.34 composite hydrogel fiber (20 wt% of SWCNTs) show respective phase transitions at around −28.6 and −33.3 °C (Supplementary Fig. 38a), which likely correspond to the freezing points of water in the fibers. The antifreezing properties are derived from the polyelectrolyte nature of the fibers, which contain high-concentration sodium ions. Supplementary Fig. 38b presents temperature responsive resistance change as the hydrogel conductor was immersed into liquid nitrogen and recovery of the resistance as liquid nitrogen slowly evaporated.

Ultrastretchable and extremely low temperature tolerable LED lighting is unraveled in Fig. 5f. No conspicuous brightness degradation was noticed up to 1200% strain at ambient temperature. The helical hydrogel conductor retained its conductivity and stretchability at −30 °C. Notably, even though the hydrogel conductor froze in liquid nitrogen (−196 °C), it could rapidly recover its mechanical stretchability within 5 s upon exposure to ambient air (Supplementary Movie 10). Besides, the helical hydrogel conductor is suited for long-term usage, considering an ultrastretchable conductor could still light up LEDs after being stored at ambient for 5 months (Supplementary Fig. 39).

**Large strain capable thermoelectric Janus hydrogel spring**. TE technology has good prospects for powering wearable electronics since it directly converts low-grade heat (<100 °C) into continuous and stable power output. However, TE devices containing hard component units or flexible TE devices usually have either no or low stretchability[44–47], which limits their adaptability to soft and curved surfaces. Meanwhile, intrinsically stretchable TE materials are facing the problem of performance degeneration under high strain owing to the prevalent increase in resistance issue[48,49]. Here, all-soft, stretchable hydrogel springs and Janus hydrogel springs are conceptualized as large-strain capable TE

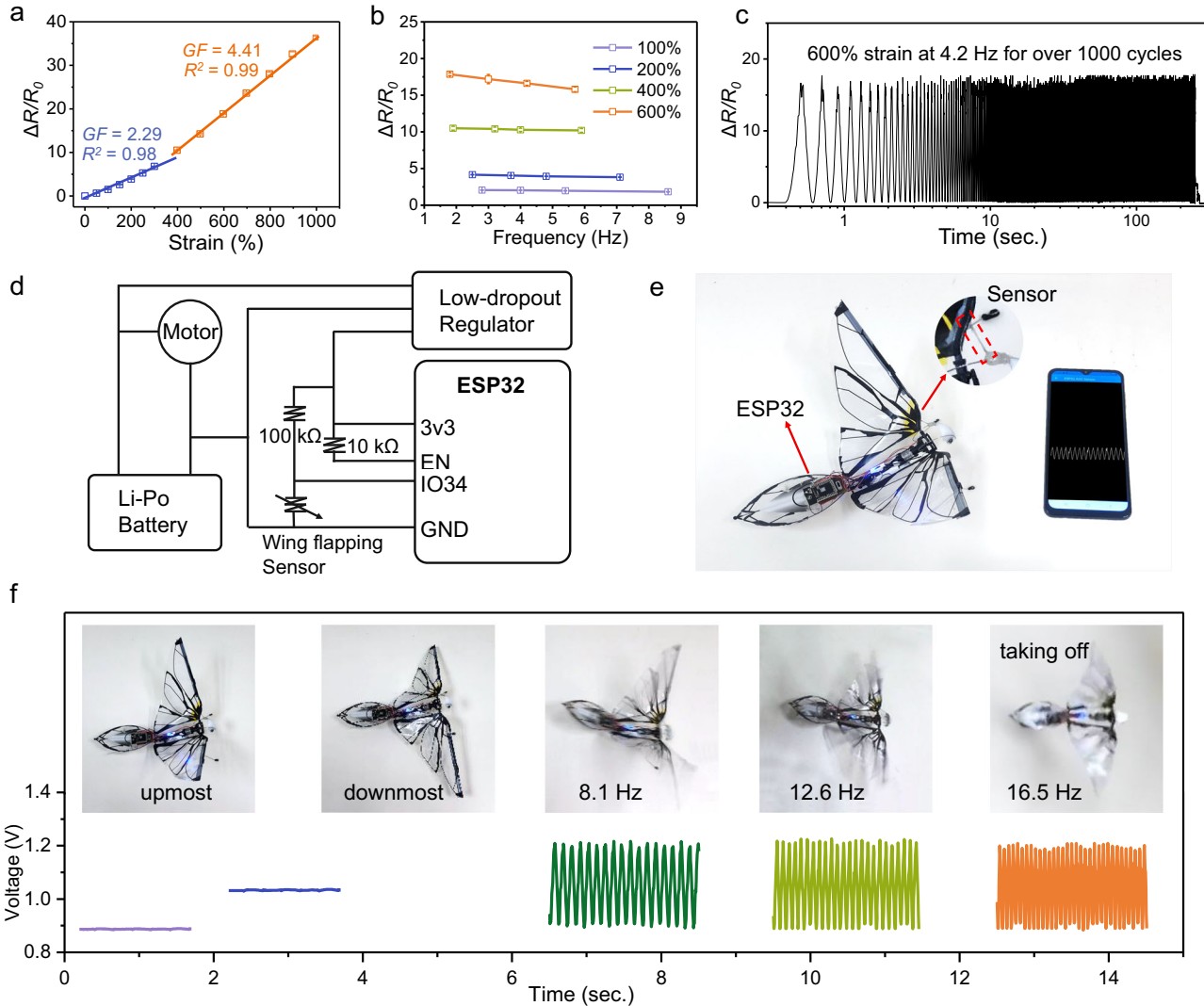

**Fig. 4 Fast resilient ionic hydrogel fiber sensors monitoring large-strain, high-speed motions. a** Resistance change of the hydrogel fiber versus tensile strain, and the gauge factor (*GF*). $R^2$ represents the coefficient of determination of the fitting. **b** Resistance change versus stretching frequency at different strains. **c** Resistance change during cyclic stretching at 600% strain and 4.2 Hz for over 1000 cycles. **d** Circuit scheme of the fiber sensor for wirelessly monitoring wing flapping of a robotic bird. **e** Photograph of the robotic bird installed with the hydrogel fiber sensor, bluetooth chip and power source. **f** Wirelessly received sensing signals corresponding to different motion states of the robotic bird. Error bars represent SD.

generators with constant power output (Supplementary Fig. 40). The hydrogel spring diameter of ~3 mm, and conductivity of 400.6 S m$^{-1}$ (pH 13.34 hydrogel doped by 40 wt% SWCNTs) was programmed.

As the hydrogel spring was stretched and straightened out, the output open circuit voltages ($V_{oc}$), electrical resistance and Seebeck coefficient were stable in the range of 0–1000% strain, suggesting the reliable TE performance (Supplementary Fig. 41). Accordingly, the Janus hydrogel springs consisting of in-series connected TE coils achieve an amplified power output. Figure 6a and Supplementary Fig. 42 show the photographs and infrared images of the Janus spring consisting of 11 TE coils, at 0–1000% strain and hot substrate-air temperature difference of 40 °C. At 0% strain, heat conduction across the adjacent contacted coils leads to a small temperature difference between the bottom and top electrodes; at 100–700% strain, the average temperature difference is stable at ~17 °C; at 800–1000% strain, the temperature gradient reduces due to the straightened coils. As a result, the generated $V_{oc}$ is stable and closely follows the temperature difference change (Fig. 6b), albeit at 100–700%

strain (Fig. 6c). The Janus spring on the hot tube of ~60 °C generates stable $V_{oc}$ of ~5.7 mV and short circuit current ($I_{sc}$) of ~2.6 μA over 5 h (Supplementary Fig. 43). As measured at 700% strain, the voltage and current increase as the temperature of the substrate increases from 5 to 40 °C higher than the ambient, and the output peak power increases from 0.1 to 5.7 nW (Fig. 6d). Furthermore, a long, all-soft Janus hydrogel spring consisting of over 100 TE coils can be readily manufactured (Fig. 6e). Figure 6f and Supplementary Fig. 44 demonstrate the utilization of the Janus TE hydrogel springs as compliant wearable and coilable bands with continuous and stable $V_{oc}$ and $I_{sc}$, i.e., $V_{oc}$ of ~11.5 mV and $I_{sc}$ of ~0.5 μA (body), and $V_{oc}$ of ~65.7 mV and $I_{sc}$ of ~2.7 μA (hot tube) (Fig. 6g).

## Discussion

We have developed a polyelectrolyte macromolecule conformational shaping method for extreme mechanical programming of polymorphic hydrogel fibers. Through controlled hydrogel filamentation based on pH-dependent antisolvent phase separation,

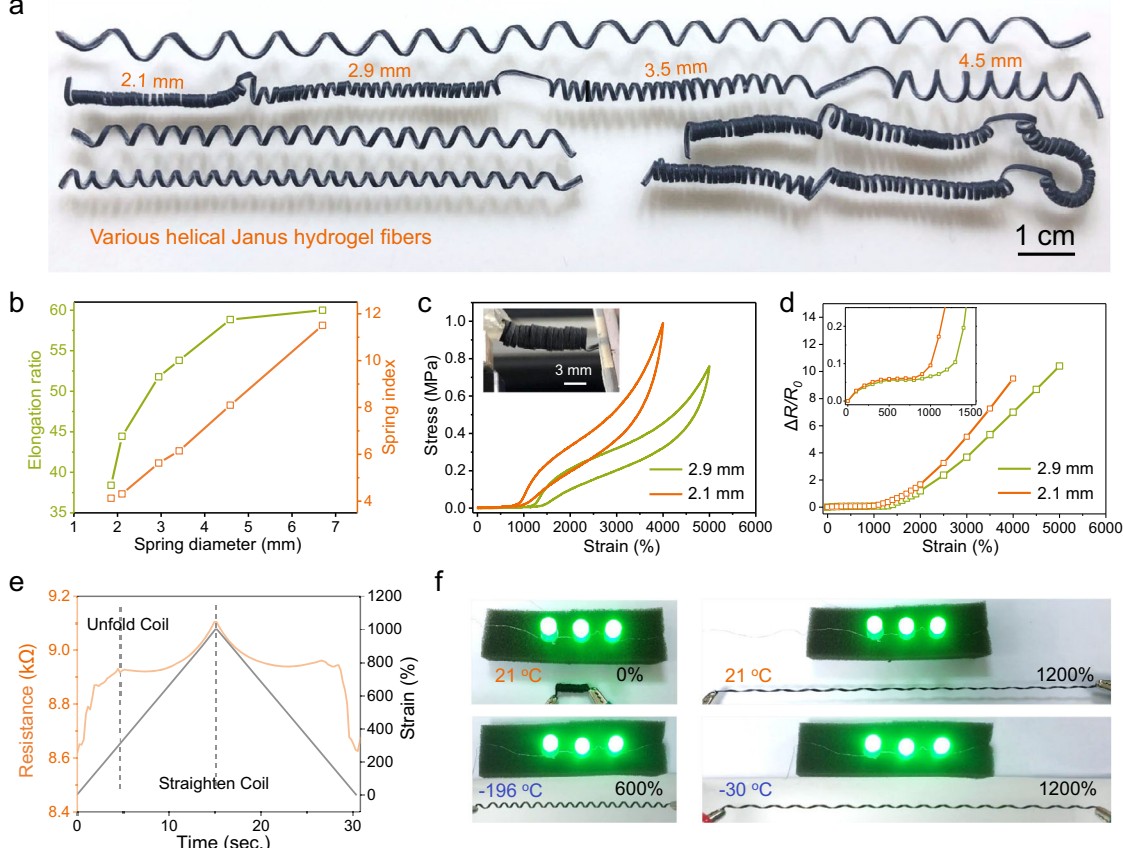

**Fig. 5 Helical Janus hydrogel fiber conductors of extremely large deformations and low temperatures tolerance. a** Photograph of various helical Janus fibers. **b** Elongation and spring index versus the spring diameter. **c** Cycled loading of the fiber conductors to large strains. The inset shows the helical fiber consisting of intimately contacted coils of ~2.9 mm diameter. **d** Resistance change of the conductors as a function of tensile strain. **e** Dynamic resistance of the conductor (~2.9 mm diameter) under a 1000% strain cycle. **f** Ultrastretchable and low temperatures tolerable LED lighting.

conformations of the polyacrylic acid based polyelectrolyte macromolecule from compactly coiled to extended, aligned states are delicately controlled. We have revealed the evolving macromolecule architectures of the single-composition hydrogel microfibers that yield extremely wide mechanical specifics, ranging from softness to ultrahigh rigidity (modulus of $240 \pm 30$ kPa to $2050 \pm 370$ MPa), brittleness to ultrastretchability (elongation of $105\% \pm 2\%$ to $2630\% \pm 120\%$), and plasticity to anelasticity and elasticity. The macromolecule conformation varied hydrogel microfibers can be continually built up in layered formats, to enable mechanical programming for polymorphic and robust hydrogel devices. Such a device fabrication method addresses the methodological gap of advanced hydrogel systems that demand integration of dissimilar or even mismatched properties for diverse biomechanical and electronic properties, besides fulfilling geometry features, without compromising interfacial integrity and process simplicity. We demonstrate the programming of ultra-large strain capable, all-soft hydrogel devices that operate durably in ambient air: 1000% strain and fast response (~30 ms) fiber sensors monitoring robotic bird dynamics, extremely large deformations (3800 to 6000%) and antifreezing helical conductors, and wearable, stretchable (700%) TE Janus springs. Our method for mechanical programming of hydrogels is not only limited to manufacturing of presented fiber electronic devices in this work, but also can extend to other functional hydrogel devices that require purpose-built integration of heterogeneous units. It represents an exclusive material and engineering strategy to date that achieves mechanical programming of multifunctional hydrogel fiber devices, therefore offering a powerful manufacturing platform for realizing realistic applications

of hydrogel devices with tunable mechanical property, adaptive form factors, high integration density and multifunctionality.

## Methods

**Preparation of hydrogel dopes**. The polyelectrolyte spinning dopes were prepared as follows: 0.73 g of sodium polyacrylate (PANa, Mw ~$3 \times 10^7$ Da, Sinopharm Chemical Reagent Co. Ltd.) was slowly added to 20 ml of deionized water, and stirred, heated at 80 °C for 5 h to get a uniform and transparent hydrogel. The obtained viscous hydrogels were centrifuged at $400 \times g$ for 30 to 60 min until the bubbles were all removed. The dopes of pH ranging from 3.00 to 9.14 were prepared by using HCl solutions of 1.00 to 0.01 mol/l. For example, the dopes of pH 9.14 and pH 3.95 were formed by dissolving 0.73 g of PANa respectively in 20 ml of 0.01 and 0.25 mol/l HCl solutions. The dope solutions of pH in the range of 12.38 to 13.97 were prepared by using NaOH solutions of 0.2 to 1.0 mol/l. The spinning dopes for pH 13.34 composite hydrogel fibers with different contents (10 wt%, 20 wt%, 30 wt% and 40 wt% relative to the PANa polymer) of SWCNTs (Carbon Solutions, Inc., AP-SWCNT) were prepared as follows: 0.073 to 0.292 g of SWCNTs with sodium dodecyl sulfate of 30 wt % relative to SWCNTs were dispersed in 20 ml of NaOH solution (0.5 mol/l), followed by 200 W 20 kHz ultrasonication treatment of 10 s on/off cycles for 30 min under ice water bath. Then 0.73 g of PANa was dissolved in the treated SWCNT solutions to get the composite hydrogel dopes.

**Fabrication of polymorphic hydrogel fibers**. To spin polyelectrolyte hydrogel fibers, the dope solutions were extruded into a methanol bath at a constant speed of 0.15 ml/min using a nozzle with the inner diameter of 900 μm with set-up placed in a fumehood. The as-spun long fibers were fixed onto a roller and collected at a speed that did not exert tension to the polyelectrolyte hydrogel fibers. The fibers had undergone phase transition in the methanol bath and subsequently solidified and straightened fibers were exposed to the ambient air (65% humidity) to produce stabilized hydrogel fibers. Preparation of highly stretchable and resilient pH 12.38 hydrogel ribbons were demonstrated by utilizing a flat nozzle with a dimension of 10 mm × 0.7 mm. To fabricate Janus hydrogel fibers, a spinneret consisting of two parallel nozzles with the dimension of 3 mm × 0.5 mm was used, the pH 12.38 hydrogel and pH 13.34 composite hydrogel flowed in parallel at 0.15 ml/min. The coaxial spinneret (inner

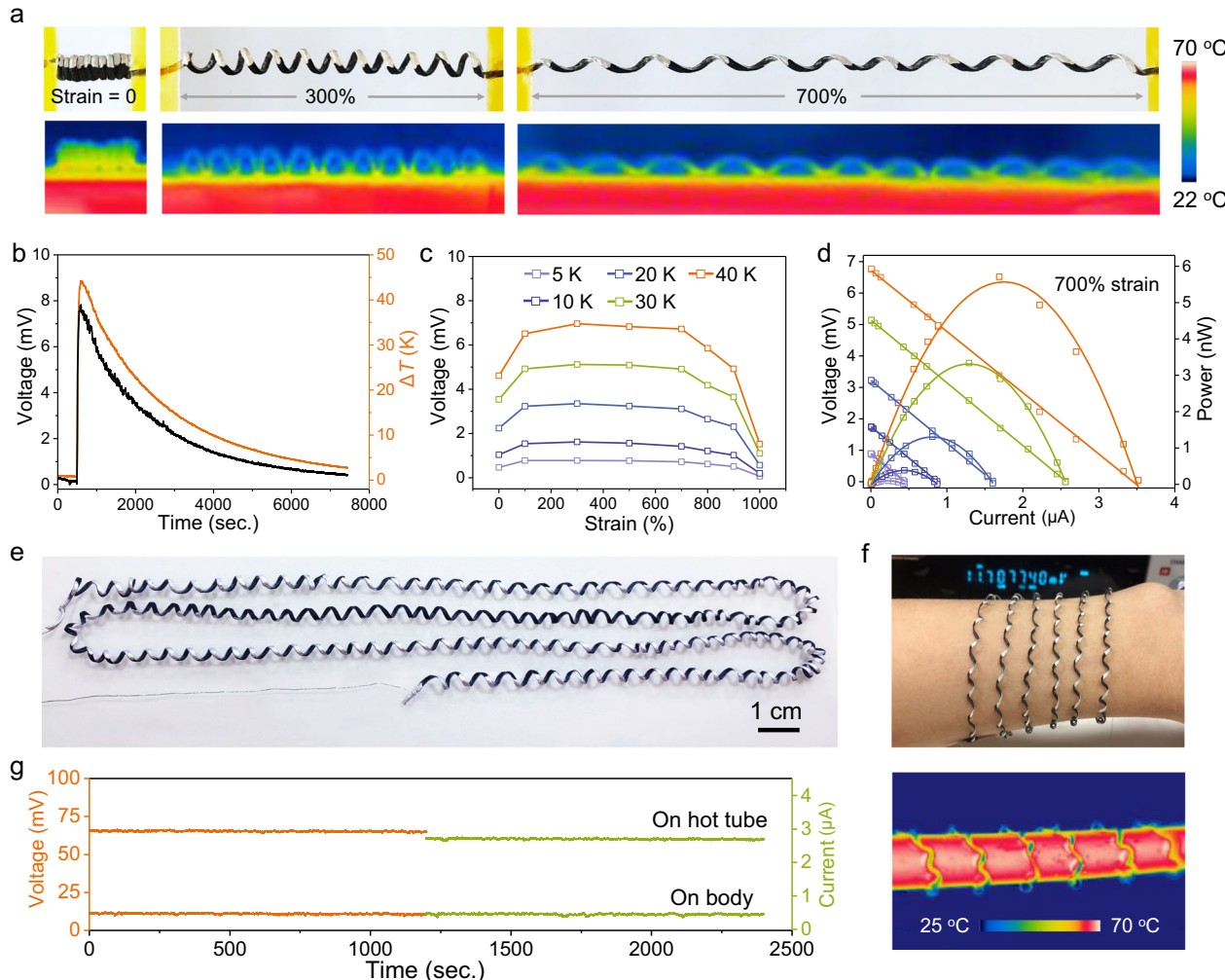

**Fig. 6 Utilization of Janus hydrogel spring as a stretchable and compliant thermal energy harvester. a** Photographs and infrared images of the Janus spring at different strains and a temperature difference of 40 °C between the hot substrate and ambient air. **b** $V_{oc}$ of the Janus spring varies the temperature difference over with time. **c** $V_{oc}$ at different strains and temperature difference. **d** TE performance at 700% strain of different temperature difference. **e** Photograph of a Janus hydrogel spring composed of >100 TE coils electrically connected in series. **f** Photograph of a wearable TE bracelet (top) and infrared image of a coilable TE (bottom). **g** $V_{oc}$ and $I_{sc}$ of the TE bracelet and coil.

diameters of 330 μm and 1.8 mm) was used to produce core-shell hydrogel fibers. The helical Janus hydrogel fibers with different diameters were fabricated by stretching the Janus fibers to different strains at a speed of 40 mm/min and then the strains were released. The content of SWCNTs in the pH 13.34 composite hydrogel phase was 20 wt % relative to the PANa polymers. To prepare TE Janus hydrogel springs, the content of SWCNTs in the pH 13.34 composite hydrogel phase was increased to 40 wt% to enhance electrical conductivity. And helical hydrogels with diameter of ~3 mm were prepared by applying a 100% prestrain to the Janus fiber. Then, Janus hydrogel spring consisted of TE coils that were electrically connected in series.

## Data availability

The data that support the findings of this study are available within the article and Supplementary Information files, or available from the corresponding authors on request. Source data are provided with this paper.

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

## Acknowledgements

The research is supported by A*STAR under its 2019 AME IRG & YIRG Grant Calls (A2083c0059, G.W.H.) and the Advanced Research and Technology Innovation Centre (ARTIC), the National University of Singapore under Grant (R261-518-014-720, G.W.H.).

## Author contributions

G.W.H. proposed the research direction and supervised the project. X.Q.W. designed the experiments and analyzed the data. K.H.C. designed the wireless motion monitoring robotic bird. W.L. conducted AFM characterization. T.D. contributed to the thermo-electric hydrogel fiber design and data analysis. S.W.L.N. and M.H. conducted the confocal microscope characterization and data analysis. Y.C. and T.L. contributed to the data analysis. G.W.H., X.Q.W. and B.C.K.T. wrote and revised the manuscript. All authors participated in the discussion and reviewed the manuscript before submission.

## Competing interests

The authors declare no competing interests.
