## [Peer Review File · Nature Communications]

Macromolecule conformational shaping for extreme mechanical programming of polymorphic hydrogel fibersReviewers' Comments:

Reviewer #1:

Remarks to the Author:

The manuscript "Macromolecule conformational shaping for extreme mechanical programming of polymorphic hydrogel fibers" suggested a macromolecule conformational shaping strategy for mechanical programming of polymorphic hydrogel fiber-based devices. Even this paper presented interesting hydrogels with extremely mechanical tunability and their utilization for extraordinary electronic devices, there are several following issues need to be solved by authors to be published. This article is worth publishing after minor revision.

1. The authors suggested that the hydrogel microfibers can lock water within the polyelectrolyte polymer networks in ambient air. In general, water in the hydrogel microfiber can move through diffusion. Therefore, it would be better to suggest additional explanation as to why the water is being trapped in the network.
2. The authors presented the hydrogels of varied macromolecular architectures, and complex shapes of polymorphic hydrogel fibers. The reviewer just wonders how the hydrogel layers are attached and the exchange of different pH solvent at the interface does not occur.
3. The authors used the Janus hydrogel spring as a stretchable and compliant thermal energy harvester. It was possible to generate stable electric energy even at 70 °C. It would be better to show that the device can operate for a long time at relatively high temperature.
4. The author fabricated the large strain and ultrafast responsive ionic hydrogel fiber sensors. Did the authors add some ions to realize the sensor with excellent characteristics? If so, do the properties of sensor change depending on the type of ions?

Reviewer #2:

Remarks to the Author:

The study consider how fibril-forming hydrogels can be utilized for extreme shape control and tailoring mechanical properties i a wide range. Overall, the paper can be published after some improvements during major revision as indicated below:

- novelty and difference with current research should be highlighted and analyzed because a lot of examples of hydrogel synthesis and shape changes fabrication;
- introduction of their materials is confusing and must be extended-not clear what is so special about PAN materials that makes them unique. show chemical schematics, interactions, and reorganizations which will illustrate clearly the uniqueness of the materials and approaches;
- along the same line- unique comprehensive central schematic that introduce components, materials and their modifications like Janus, CNT composite better to be integrated and introduced in the very beginning;current fig. 1 does not cover all options and does not show novelty with fine details;
- SAXS data analysis is too simplistic: discuss more, derive dimensions, calculate orientation parameters to support conclusions;
- larger scale AFM should be shown to characterize morphology on micron scale, not just nanoscale;
- what is swelling ration and what happens with overall bulk density during fibrillization;
- CNT-based composites: what happens with CNT component, it distribution, just stating "doping with 40%" is not enough etc etc
- through the paper: all characteristics are presented with overestimated precision, e.g. strain reaching 2693.0%; elongation changes from 108.3%, etc etc. The authors should work thoroughly, do meaningful rounding, provide STD and statistics;
- mechanical performance down to -193C is overstated- fibers must be heated above -30C to perform in the end, changes in representation of true pattern should be made;
- it looks like some of interesting data from SI should be moved to the main text to facilitate discussion and represent numerous illustrative videos;

- referencing is reasonable but some basic review articles of responsive polymers should be added (e.g., Nat. Mater., 2010,9, 101-113)

REVIEWER COMMENTS

Reviewer #1 (Remarks to the Author):

The manuscript “Macromolecule conformational shaping for extreme mechanical programming of polymorphic hydrogel fibers” suggested a macromolecule conformational shaping strategy for mechanical programming of polymorphic hydrogel fiber-based devices. Even this paper presented interesting hydrogels with extremely mechanical tunability and their utilization for extraordinary electronic devices, there are several following issues need to be solved by authors to be published. This article is worth publishing after minor revision.

1. The authors suggested that the hydrogel microfibers can lock water within the polyelectrolyte polymer networks in ambient air. In general, water in the hydrogel microfiber can move through diffusion. Therefore, it would be better to suggest additional explanation as to why the water is being trapped in the network.
2. The authors presented the hydrogels of varied macromolecular architectures, and complex shapes of polymorphic hydrogel fibers. The reviewer just wonders how the hydrogel layers are attached and the exchange of different pH solvent at the interface does not occur.
3. The authors used the Janus hydrogel spring as a stretchable and compliant thermal energy harvester. It was possible to generate stable electric energy even at 70 °C. It would be better to show that the device can operate for a long time at relatively high temperature.
4. The author fabricated the large strain and ultrafast responsive ionic hydrogel fiber sensors. Did the authors add some ions to realize the sensor with excellent characteristics? If so, do the properties of sensor change depending on the type of ions?

Reviewer #2 (Remarks to the Author):

The study consider how fibril-forming hydrogels can be utilized for extreme shape control and tailoring mechanical properties i a wide range. Overall, the paper can be published after some improvements during major revision as indicated below:

- novelty and difference with current research should be highlighted and analyzed because a lot of examples of hydrogel synthesis and shape changes fabrication;
- introduction of their materials is confusing and must be extended-not clear what is so special about PAN materials that makes them unique. show chemical schematics, interactions, and reorganizations which will illustrate clearly the uniqueness of the materials and approaches;
- along the same line- unique comprehensive central schematic that introduce components, materials and their modifications like Janus, CNT composite better to be integrated and introduced in the very beginning; current fig. 1 does not cover all options and does not show novelty with fine details;
- SAXS data analysis is too simplistic: discuss more, derive dimensions, calculate orientation parameters to support conclusions;

- larger scale AFM should be shown to characterize morphology on micron scale, not just nanoscale;
- what is swelling ration and what happens with overall bulk density during fibrillization;
- CNT-based composites: what happens with CNT component, it distribution, just stating "doping with 40%" is not enough etc etc
- through the paper: all characteristics are presented with overestimated precision, e.g. strain reaching 2693.0%; elongation changes from 108.3%, etc et.0.0.0.0c. The authors should work thoroughly, do meaningful rounding, provide STD and statistics;
- mechanical performance down to -193C is overstated- fibers must be heated above -30C to perform in the end, changes in representation of true pattern should be made;
- it looks like some of interesting data from SI should be moved to the main text to facilitate discussion and represent numerous illustrative videos;
- referencing is reasonable but some basic review articles of responsive polymers should be added (e.g., Nat. Mater., 2010,9, 101-113)

Response to reviewer comments for manuscript:

“Macromolecule conformational shaping for extreme mechanical programming of polymorphic hydrogel fibers” (NCOMMS-21-51572-T)

Reviewer #1 (Remarks to the Author):

Reviewer: The manuscript “Macromolecule conformational shaping for extreme mechanical programming of polymorphic hydrogel fibers” suggested a macromolecule conformational shaping strategy for mechanical programming of polymorphic hydrogel fiber-based devices. Even this paper presented interesting hydrogels with extremely mechanical tunability and their utilization for extraordinary electronic devices, there are several following issues need to be solved by authors to be published. This article is worth publishing after minor revision.

Response: We are thankful to the reviewer for his/her positive assessment of our work, and we have revised the manuscript as suggested by the reviewer.

Reviewer: The authors suggested that the hydrogel microfibers can lock water within the polyelectrolyte polymer networks in ambient air. In general, water in the hydrogel microfiber can move through diffusion. Therefore, it would be better to suggest additional explanation as to why the water is being trapped in the network.

Response: The sodium polyacrylate polyelectrolyte (PANA) with abundant carboxylate groups is hygroscopic, the polyelectrolyte hydrogel microfibers are capable of stable water retention in the ambient air (65% humidity) without the need of adding additional moisturizing factors. Meanwhile, the equilibrated water contents of the hydrogel microfibers depend on conformations of the PANA polyelectrolyte macromolecules. As suggested by the reviewer, we have added an additional explanation to clarify the property of the polyelectrolyte polymer in the revised manuscript.

“We exploit the conformational tuning³⁷ of the polyelectrolyte macromolecule based on a pH-dependent antisolvent phase separation induced hydrogel filamentation process. The poly(acrylic acid) based polyelectrolyte, commercially available sodium polyacrylate (PANA) of molecular weight ($\sim 3 \times 10^7$ Da) was chosen as the starting material. Especially, the high molecular weight of PANA is required for polymer networks with entangled long macromolecule chains, which are promising for acquiring wide-range tunable mechanical properties when the coiling state of the macromolecule chains changes. In addition, PANA polyelectrolyte with abundant carboxylate groups is hygroscopic, capable of stable water retention in the ambient air (65% humidity).” (P4, line 10-18)

“Importantly, those microfibers regain water from the ambient air due to the hygroscopic nature of PANA, and their equilibrated water contents depend on conformations of the polyelectrolyte macromolecules, which vary from ~ 12 to 40 wt% with increasing pH (Supplementary Figs. 4,5).” (P5, line 28-30)

Reviewer: The authors presented the hydrogels of varied macromolecular architectures, and complex shapes of polymorphic hydrogel fibers. The reviewer just wonders how the hydrogel layers are attached and the exchange of different pH solvent at the interface does not occur.

Response: In this revised manuscript, we have added Supplementary Video 6 and Supplementary Figure 27 to show the formation process of Janus hydrogel fiber. The changes in the main text and Supplementary information are as follows:

“Supplementary Video 5 presents continuous fabrication of Janus hydrogel fibers, and conformally bonded interface forms through the synchronous phase separation induced filamentation of the two phases and the spontaneous entanglement of macromolecules at the interface (Fig. 3c). Supplementary Fig. 27 and Video 6 show the formation process of well-bonded interfaces at different flow rates.” (P10, line 12-16)

Supplementary Figure 27. (a) Formation of well-bonded interfaces in the Janus hydrogel fibers at flow rates of 0.15 mL/min and (b) 0.025 mL/min.

“Upon extrusion into the methanol bath, the two phases were subjected to antisolvent phase separation induced dehydration process. In the beginning, we observed that the interface partially bonded with gaps, and the water in the two hydrogel phases came out from around the Janus structure. With the progress of phase separation, the Janus structure continuously shrank, and the two hydrogel dopes at the interface were forced to fully contact and bond together in the following dehydration process. The complete process was recorded by the microscope at a low flow rate of 0.025 mL/min as shown in the Supplementary Video 6.” (P33 in Supplementary materials)

Reviewer: The authors used the Janus hydrogel spring as a stretchable and compliant thermal energy harvester. It was possible to generate stable electric energy even at 70 °C. It would be better to show that the device can operate for a long time at relatively high temperature.

Response: By utilizing the thermoelectric property of the single-walled carbon nanotubes, the Janus hydrogel spring can maintain stable thermoelectric performance at the relatively high temperature. To demonstrate the stability, we attached the Janus hydrogel spring on the hot tube filled with circulating water of ~90 °C and measured the open circuit voltage and short circuit current. The results are now shown in Supplementary Figure 43.

“The Janus spring on the hot tube of ~60 °C generates stable V_{oc} of ~5.7 mV and short circuit current (I_{sc}) of ~2.6 μ A over 5 hours (Supplementary Fig. 43).” (P16, line 15-17)

Supplementary Figure 43. V_{oc} and I_{sc} generated by the 500% strain Janus spring on the hot tube filled with circulating water of ~90 °C.

“The surface temperature of the hot tube was ~60 °C, and the Janus spring consisting of 11 TE coils was exposed to hot substrate-air temperature difference (ΔT) of ~36 °C. The generated V_{oc} of ~5.7 mV and I_{sc} of ~2.6 μ A were stable over 5 hours.” (P49 in Supplementary materials)

Reviewer: The author fabricated the large strain and ultrafast responsive ionic hydrogel fiber sensors. Did the authors add some ions to realize the sensor with excellent characteristics? If so, do the properties of sensor change depending on the type of ions ?

Response: In this work, we show the macromolecule conformation engineering of the PANa polyelectrolyte for mechanical property modulation of the hydrogel microfibers. The PANa

polyelectrolyte hydrogel itself is conductive due to the mobile sodium ions inside. Actually, we only tuned the pH of PANa polyelectrolyte hydrogel dopes using HCl and NaOH solutions, to change the conformation of PANa macromolecule. The as-prepared pH 12.38 hydrogel microfiber was readily used as the large strain and ultrafast responsive ionic hydrogel sensor owing to the high stretchability, elasticity and ionic conductivity. In the revised manuscript, we have modified Fig. 1 to clearly illustrate the chemical structure of the polyelectrolyte hydrogel.

Fig. 1 Extreme mechanical property tuning and shaping of polymorphic hydrogel fibers. a Conformational shaping of the polyelectrolyte macromolecule based on the pH-dependent antisolvent phase separation to generate hydrogel microfibers of complex network architectures (left), and their microfibrillar units coupling to produce polymorphic hydrogel fibers (right). **b** Mechanical parameters scope of the hydrogel microfibers with pH from 3.95 to 13.97. **c** Mechanical gradation of the microfibers with increased pH. (P4)

Reviewer #2 (Remarks to the Author):

Reviewer: *The study consider how fibril-forming hydrogels can be utilized for extreme shape control and tailoring mechanical properties i a wide range. Overall, the paper can be published after some improvements during major revision as indicated below:*

Response: We are thankful for the reviewer's helpful suggestions for our work. Following the reviewer's advice, we have made a major revision to the manuscript. We have clarified the novelty of our materials approach, the selection of polymer materials and the details of the pH-dependent antisolvent phase separation induced hydrogel filamentation, and better characterized and analyzed the microstructure of the hydrogel microfibers.

Reviewer: *- novelty and difference with current research should be highlighted and analyzed because a lot of examples of hydrogel synthesis and shape changes fabrication;*

Response: In this work, we explore the conformation engineering of PANa polyelectrolyte macromolecule based on the pH dependent antisolvent phase separation. Actually, the resulting hydrogel microfibers represent a new type of unconventional polymer networks based on the single-composition entangled polyelectrolyte macromolecule with controllable conformations. This simple, scalable approach realizes previously unachieved mechanical property modulation of a single-composition polymer hydrogel, ranging from softness to ultrahigh rigidity, brittleness to ultrastretchability, and plasticity to anelasticity and elasticity. As suggested by the reviewer, we give more details in the introduction of the background and our materials approach. The corresponding descriptions in the revised manuscript now read:

“For one thing, the conventional polymer networks of hydrogels are composed of randomly cross-linked polymer chains with covalent bonds, in which entanglements and physical cross-links of the polymer chains are negligible. The formed hydrogels are usually fragile, and the integrated devices suffer low mechanical robustness and low stretchability. For another, the unconventional polymer networks of hydrogels in terms of network architectures and interactions among polymer chains have been widely explored in the past two decades¹⁸. Representative examples include interpenetrating polymer networks based on covalent and physical hybrid cross-linking strategies¹⁹⁻²¹, polymer networks with slide-ring crosslinkers²², and polymer networks with high-functionality cross-linkers, such as crystalline domains^{23, 24}, hydrogen bonds²⁵, ionic bonds²⁶ and nano-/microparticles²⁷⁻³⁰. Remarkable mechanical properties, such as high toughness, strength, resilience and interfacial toughness have been achieved. However, these sophisticated polymer networks usually need meticulous design of the composition components, sequential or prolonged polymerization, or cumbersome pre/post-polymerization treatment. The fabrications mostly depend on custom-designed molds, which are of low throughput, limited in shape control and difficult in upscaling. So far, few material strategies have been reported that allow printing^{31, 32} or spinning^{33, 34} of mechanically robust hydrogels.” (P2, line 8-26)

“Our method shows that the conformation of a poly(acrylic acid) based polyelectrolyte macromolecule can be synthetically engineered from compactly coiled to extended, aligned states based on a pH-dependent antisolvent phase separation process (left of Fig. 1a). The pH determines

the original conformations of the polyelectrolyte macromolecules, while the phase separation drives aggregation of the macromolecules and generates densely entangled macromolecule networks. The resulting hydrogel microfibers represent a new type of unconventional polymer networks based on the single-composition entangled polyelectrolyte macromolecule with controllable conformations. And tunable mechanical properties ranging from softness to ultrahigh rigidity, brittleness to ultrastretchability, and plasticity to anelasticity and elasticity, can be programmed (Fig. 1b,c).” (P3, line 6-16)

Reviewer: - introduction of their materials is confusing and must be extended-not clear what is so special about PANa materials that makes them unique. show chemical schematics, interactions, and reorganizations which will illustrate clearly the uniqueness of the materials and approaches;

Response: We agree with the reviewer’s suggestions that the structure of the materials and the uniqueness of the materials approach, should be clearly clarified in the main text. As suggested, we have revised the introduction part, Fig. 1 and corresponding descriptions. The rationale of the polymer materials selection is explained, the structure of the polyelectrolyte polymer, and the mechanism for the pH-dependent antisolvent phase separation are schematically illustrated in Fig. 1.

Fig. 1 Extreme mechanical property tuning and shaping of polymorphic hydrogel fibers. a Conformational shaping of the polyelectrolyte macromolecule based on the pH-dependent

antisolvent phase separation to generate hydrogel microfibers of complex network architectures (left), and their units coupling to produce polymorphic hydrogel fibers (right). **b** Mechanical parameters scope of the hydrogel microfibers with pH from 3.95 to 13.97. **c** Mechanical gradation of the microfibers with increased pH. (P4)

“The pH determines the original conformations of the polyelectrolyte macromolecules, while the phase separation drives aggregation of the macromolecules and generates densely entangled macromolecule networks. The resulting hydrogel microfibers represent a new type of unconventional polymer networks based on the single-composition entangled polyelectrolyte macromolecule with controllable conformations.” (P3, line 9-14)

“We exploit conformational tuning³⁷ of the polyelectrolyte macromolecule based on a pH-dependent antisolvent phase separation induced hydrogel filamentation process. The poly(acrylic acid) based polyelectrolyte, commercially available sodium polyacrylate (PANa) of molecular weight ($\sim 3 \times 10^7$ Da) was chosen as the starting material. Especially, the high molecular weight of PANa is required for polymer networks with entangled long macromolecule chains, which are promising for acquiring wide-range tunable mechanical properties when the coiling state of the macromolecule chains changes. In addition, PANa polyelectrolyte with abundant carboxylate groups is hygroscopic, capable of stable water retention in the ambient air (65% humidity).” (P4, line 10-18)

“Upon extrusion into methanol bath, the polyelectrolyte macromolecules undergo fast dehydration as water molecules in the network are captured by methanol due to the higher affinity of methanol to water than that of carboxylate groups (left of Fig. 1a). Supplementary Video 1 presents the continuous process of hydrogel filamentation and collection of microfibers. During the dehydration and phase separation, the macromolecules of different pH aggregate and experience specific conformation changes depending on their own water affinity. The dehydration process of those pH varied hydrogel dopes takes 150 to 250 s, and all of the water in the dopes diffuses into the methanol bath and the microfibers become rigid upon the completion of the phase separation. Dopes of the lowest pH 3.95 can be solidified fastest (Supplementary Video 2), due to the weakest bonding of protonated polyelectrolyte macromolecules to water molecules. Meanwhile, the phase separation induced aggregation of the macromolecule chains generates densely entangled macromolecule networks, thereby engendering material toughness.” (P5, line 11-24)

Reviewer:- along the same line- unique comprehensive central schematic that introduce components, materials and their modifications like Janus, CNT composite better to be integrated and introduced in the very beginning; current fig. 1 does not cover all options and does not show novelty with fine details;

Response: The key point of our work is the polyelectrolyte macromolecule conformation engineering approach for mechanical property modulation and shape control of the hydrogel fibers. Meanwhile, the multi-layered hydrogel fiber containing hydrogel microfiber units of different macromolecule conformations, can be fabricated in one-step by utilizing customized spinneret. For example, we fabricated the Janus hydrogel fiber, in which the fast resilient pH 12.38 hydrogel phase serves as the elastic substrate while the counterpart ultrastretchable, plastic pH 13.34 hydrogel phase is doped with single-walled carbon nanotubes (SWCNTs). The mismatched mechanical resilience property of the two hydrogel microfiber units allows programming of diameter controllable spring-like structures by applying prestrains. Actually, good compatibility of the hydrogel matrix makes it capable of encompassing a variety of active fillers to enable functional heterogeneity. Here, SWCNTs are utilized as the functional fillers to produce high electrical conductivity and thermoelectric properties.

We agree with the reviewer's suggestions. In the revised manuscript, the macromolecule conformation control of the PANa polyelectrolyte based on the pH-dependent antisolvent phase separation, and the shape control of the hydrogel fibers based on the layering of the hydrogel microfiber units are illustrated in Fig. 1. Meanwhile, to avoid repetitive descriptions for the shape control of the hydrogel fibers in Fig.1 and Fig. 3, we present the fabrication devices and the detail mechanisms for various shape control in Fig. 3, to show the processability of the hydrogel systems, and explain the function of SWCNT fillers. The changes are as follows:

“Moreover, the heterogeneous mechanical properties resulting from the complex macromolecule architectures can be arbitrarily layered in one-step via our approach, thereby generating hydrogel fiber devices with readily customizable shapes. Polymorphic hydrogel fibers of fibers/ribbons, Janus fibers, multilayered fibers, core-shell fibers, helical Janus fibers, Janus springs and beyond, can be fabricated (right of **Fig. 1a**).” (*P3, line 16-21*)

Fig. 3 Processability of the macromolecule conformation tunable hydrogels for monolithic, polymorphic hydrogel fiber devices. **a** Schematic illustration of layered, monolithic hydrogels of varied macromolecule conformations. **b** Schematic illustration for the preparation of helical Janus hydrogel fiber (top) and fabrication of continuous Janus hydrogel fiber via a parallel-axial dual-spinneret (bottom). **c** Photograph of the Janus hydrogel fiber of >1 meter in length and optical microscope image showing the conformal interface of layered hydrogel microfibers. **d** Formation of a helical Janus fiber in response to 900% prestrain. **e** Spring diameter and number of coils of helical Janus fibers versus applied prestrain. (P11)

“Here, the polyelectrolyte macromolecules of varied conformations offer the single-composition hydrogels various mechanical properties, and meanwhile, the hydrogel matrix could encompass a variety of active fillers to enable functional heterogeneity.” (P9, line 16-19)

“A parallel-axial dual-spinneret system is devised to layer two microfibers in a stacked structure, forming Janus hydrogel fibers of arbitrary length. The fast resilient pH 12.38 hydrogel phase serves as the elastic substrate while the counterpart ultrastretchable, plastic pH 13.34 hydrogel phase is doped with single-walled carbon nanotubes (SWCNTs) to evolve heterogeneity in electrical conductivity and mechanical resilience (Fig. 3b). Here, SWCNTs are utilized as the functional fillers to produce high electrical conductivity and thermoelectric properties.” (P9-10)

Reviewer:- SAXS data analysis is too simplistic: discuss more, derive dimensions, calculate orientation parameters to support conclusions;

Response: As suggested by the reviewer, we have given more discussion on the SAXS data in the revised manuscript. To obtain orientation parameters of the orientated microfiber at different tensile strains, we did WAXS measurements, and calculated the orientation factors, and the results have been shown in Supplementary Fig. 14. In consideration of the length limit of the main text, we give the main conclusions in the main text, and detailed discussions in the supporting information. The changes are as follows:

“Small-angle X-ray scattering (SAXS) results reveal the evolution of delicate structures below 100 nm. Uniform, continuous polymer networks of low scattering contrast in the moderate pH range display weak scattering signal whereas the networks containing aggregated domains exhibit strong scattering peaks (Supplementary Fig. 13a,c). The stretched SAXS pattern perpendicular to the fiber axis of pH 13.97 microfiber signifies the presence of aligned nanostructures, as proven by the AFM results. The extension and reorientation of macromolecule chains in the pH 9.14 microfiber under stretching produces scattering centers perpendicular to the stretching direction³⁹, and the continuously increased macromolecule orientation within 800% tensile strain leads to intensified and narrowed scattering in the perpendicular direction (Supplementary Fig. 13b,d-f). We used Wide-angle X-ray scattering (WAXS) measurements to determine the macromolecule orientation parameters at different strains (Supplementary Fig. 14), and the calculated orientation factor (f) is 0.42 at 0% strain, which increases to 0.67 at 100% strain, 0.80 at 400% strain, and 0.91 at 800% strain. The increase in f indicates that the orientation of macromolecule chains along the microfiber axis continuously improves with increasing stretching ratio.” (P6-7)

Supplementary Figure 13. Investigation of the delicate structures evolution below 100 nm using SAXS. (a) SAXS patterns and (c) profiles of the hydrogel microfibers of different pH. (b) SAXS patterns and (d) azimuthal angle dependent scattering profiles of pH 9.14 fiber at different strains. The white double arrows indicate the length direction of the microfibers and the gray sectors indicate the 20° region for intensity integration. (e) 1D integrated intensity curve of the pH 9.14 microfiber at 800% strain as a function of the azimuthal angle. Lorentz fitting is used to fit the peak and to obtain the full width at half maximum (FWHM). (f) FWHM as a function of strain.

“Hydrogel microfibers in the moderate pH range (pH 6.35, 9.14 and 12.38) display very weak scattering, as the uniformly entangled and continuous networks have low scattering contrast, whereas the hydrogel microfibers (pH 3.95, 13.34 and 13.97) containing aggregated nanoscale domains (polymer nanoclusters or crystalline domains) display high scattering intensity. The strongest diffraction peak at $q = 0.01 \text{ \AA}^{-1}$ in the SAXS spectra suggests the presence of aggregated domains with a mean interdomain spacing of $\sim 63 \text{ nm}$ (Supplementary Fig. 13a,c). It should be noted that the lab-source SAXS instrument can only detect the signals in the polymer network where relatively high scattering contrasts are present. Meanwhile, the stretched SAXS pattern of pH 13.97 microfiber indicates the strong scattering perpendicular to the microfiber axis, which is attributed to the aligned nanostructures as shown by the AFM results.

Upon being stretched, the scattering intensity of pH 9.14 hydrogel microfiber becomes stronger perpendicular to the stretching direction, suggesting the appearance of scattering centers induced by the macromolecule reorientation (Supplementary Fig. 13b). We computed FWHM from the azimuthal scan profile to characterize the macromolecule orientation under different strains (Supplementary Fig. 13d-f). FWHM = 180° means an isotropic distribution of the nanonetworks, while FWHM = 0° means a perfectly anisotropic distribution¹. FWHM is ~110° at 0% strain, and decreases to ~43° at 100% and ~11° at 800% strain.” (P18-19 in Supplementary materials)

Supplementary Figure 14. Orientation factors of the pH 9.14 microfiber at different strains. (a) WAXS patterns of the microfiber at different strains. The grey arrows indicate the length direction of the microfibers. (b) Azimuthal scanning at $q = 0.58 \text{ \AA}^{-1}$ and Lorentz fitting results.

“Stretching of the pH 9.14 microfiber induces extension and reorientation of the macromolecule chains. Before stretching, a nearly homogenous diffraction ring is observed in the WAXS pattern of the microfiber (Supplementary Fig. 14a). The stretching promotes macromolecule orientation along the microfiber direction, which yields a strong angle dependence in its diffraction pattern with high-intensity arc areas perpendicular to the microfiber axis at 100% and 400% strain. High-intensity diffraction spots are shown in the microfiber at 800% strain. The corresponding integrated intensity scans as a function of the azimuthal angle are shown in Supplementary Fig. 14b. The orientation factor, f is determined by the following equation:

$$f = \frac{180^\circ - \Delta\phi_{1/2}}{180^\circ}$$

where $\Delta\phi_{1/2}$ represents FWHM of the azimuthally scanned peak². The value of f is 1 when the polymer chains align perfectly parallel to the fiber direction, and when f is zero, it means there is random orientation. The microfiber at 0% strain gives $f = 0.42$, and $f = 0.91$ at 800% strain indicates that the macromolecule chains are oriented along the microfiber direction.” (P20 in Supplementary materials)

References

1. Cui, K. et al. Multiscale energy dissipation mechanism in tough and self-healing hydrogels. *Phys. Rev. Lett.* **121**, 185501 (2018).
2. Kongkhleng, T., Tashiro, K., Kotaki, M. & Chirachanchai, S. Electrospinning as a new technique to control the crystal morphology and molecular orientation of polyoxymethylene nanofibers. *J. Am. Chem. Soc.* **130**, 15460-15466 (2008).

(P52 in Supplementary materials)

Reviewer:- larger scale AFM should be shown to characterize morphology on micron scale, not just nanoscale;

Response: As suggested by the reviewer, we have added larger scale AFM images for the hydrogel microfibers of different pH in Supplementary Figure 7.

Supplementary Figure 7. AFM images of the hydrogel microfibers of different pH. (P12 in Supplementary materials)

Reviewer:- what is swelling ration and what happens with overall bulk density during fibrillization;

Response: The water content of the hydrogel spinning dopes is fixed at 96.5 wt%. When antisolvent phase separation-induced dehydration finishes in the methanol bath, all of the water in the hydrogel dopes diffuses into methanol, and the microfibers become rigid. Upon being exposed to ambient air (65% humidity), the microfibers regain water from the air, and the water contents of the hydrogel microfibers of different pH are from ~12 to 40 wt%. In Supplementary Video 1, we show that the solidified pH 9.14 microfibers become rigid and brittle after the phase separation, and then gradually become soft in the air owing to their absorption of water from the air. We have revised the explanations of the hydrogel filamentation process, which now reads:

“The dehydration process of those pH varied hydrogel dopes takes 150 to 250 s, and all of the water in the dopes diffuses into the methanol and the microfibers become rigid upon completion of phase separation. Dopes of the lowest pH 3.95 can be solidified fastest (Supplementary Video 2), due to the weakest bonding of protonated polyelectrolyte macromolecules to water molecules. Meanwhile, the phase separation-induced aggregation of the macromolecule chains generates

densely entangled macromolecule networks, thereby engendering material toughness.” (P5, line 17-24)

“Importantly, those microfibers regain water from the ambient air due to the hygroscopic nature of PANa, and their equilibrated water contents depend on conformations of the polyelectrolyte macromolecules, which vary from ~12 to 40 wt% with increasing pH (Supplementary Figs. 4,5).” (P5, line 28-30)

Reviewer:- *CNT-based composites: what happens with CNT component, its distribution, just stating "doping with 40%" is not enough etc etc*

Response: The CNT-based composite hydrogel is the pH 13.34 hydrogel blended with SWCNTs. As suggested by the reviewer, we have added SEM images of the composite hydrogel as Supplementary Figure 24, showing the uniform distribution of SWCNTs within the polymer networks.

“Here, SWCNTs are utilized as the functional fillers to produce high electrical conductivity and thermoelectric property. Scanning electron microscope images show that SWCNTs are well dispersed within the hydrogel networks (Supplementary Fig. 24).” (P10, line 3-6)

Supplementary Figure 24. (a,b) SEM images showing the surface and (c,d) internal structure of the pH 13.34 composite hydrogel (20 wt% SWCNTs). (P30 in Supplementary materials)

Reviewer:- *through the paper: all characteristics are presented with overestimated precision, e.g. strain reaching 2693.0%; elongation changes from 108.3%, etc et.0.0.0.0c. The authors should work thoroughly, do meaningful rounding, provide STD and statistics;*

Response: As suggested by the reviewer, we have provided the data with STD on the mechanical properties of the hydrogel microfibers of different pH, and revised Supplementary Figure 16 and Table S1. The changes now read:

“The pH-dependent mechanical parameters (Supplementary Fig. 16 and Table S1), correspond to the scope of elongation ratio of $105\% \pm 2\%$ to $2627\% \pm 124\%$, tensile strength of 1.21 ± 0.12 MPa

to 47.20 ± 5.15 MPa, modulus of 0.24 ± 0.03 MPa to 2.05 ± 0.37 GPa, and toughness of 1.66 ± 1.10 MJ m⁻³ to 17.84 ± 1.59 MJ m⁻³.” (P8, line 19-23)

Supplementary Figure 16. Breaking strain, strength, elastic modulus and toughness of the hydrogel microfibers as a function of pH. Error bars represent SD. (P22 in Supplementary materials)

Table S1. Mechanical properties of the hydrogel microfibers of different pH. Data are presented as average value with standard deviation.

	Breaking strain (%)	Breaking stress (MPa)	Elastic modulus (MPa)	Toughness (MJ m ⁻³)
pH 3.95	5 ± 2	47.20 ± 5.15	2046.00 ± 370.28	1.66 ± 1.10
pH 5.12	390 ± 56	6.71 ± 0.45	32.83 ± 3.55	15.41 ± 3.30
pH 6.35	850 ± 41	1.26 ± 0.07	0.28 ± 0.02	3.57 ± 0.08
pH 9.14	1091 ± 46	1.25 ± 0.07	0.24 ± 0.03	4.78 ± 0.26
pH 12.38	1542 ± 116	1.22 ± 0.04	0.27 ± 0.03	9.69 ± 1.45
pH 13.34	2527 ± 124	1.21 ± 0.12	0.34 ± 0.03	17.84 ± 1.59
pH 13.97	594 ± 83	2.50 ± 0.18	23.30 ± 6.48	11.94 ± 1.76

(P51 in Supplementary materials)

Reviewer: - *mechanical performance down to -193C is overstated- fibers must be heated above -30C to perform in the end, changes in representation of true pattern should be made;*

Response: The helical hydrogel conductor retained its conductivity and stretchability at $-30\text{ }^{\circ}\text{C}$. Notably, even though the hydrogel conductor froze in the liquid nitrogen ($-196\text{ }^{\circ}\text{C}$), it could rapidly recover its mechanical stretchability within 5 seconds upon exposure to ambient air. We agree with the reviewer's suggestion, and have removed the previous statement in the abstract and conclusion parts in the revised manuscript, which now reads:

“Our approach yields hydrogel microfibers of varied macromolecule conformations that can be built-in layered formats, enabling the translation of extraordinary, realistic hydrogel electronic applications, i.e. large strain (1000%) and ultrafast responsive ($\sim 30\text{ ms}$) fiber sensors in a robotic bird, large deformations (6000%) and antifreezing helical electronic conductors, and large strain (700%) capable Janus springs energy harvesters in wearables.” (P1)

“We demonstrate the programming of ultra-large strain capable, all-soft hydrogel devices that operate durably in ambient air: 1000% strain and fast response ($\sim 30\text{ ms}$) fiber sensors monitoring robotic bird dynamics, extremely large deformations (3800 to 6000%) and antifreezing helical conductors, and wearable, stretchable (700%) thermoelectric Janus springs.” (P18, line 12)

Reviewer: - *it looks like some of interesting data from SI should be moved to the main text to facilitate discussion and represent numerous illustrative videos;*

Response: We agree with the reviewer's suggestions. And in consideration of the length limit of the main text, we have selectively moved some important information to the main text. In the revised manuscript, schematic illustration for the mechanism of pH-dependent antisolvent phase separation has been moved to Fig. 1a, and the schematic illustration for preparation of helical Janus hydrogel fiber and devices for the fabrication of continuous Janus hydrogel fiber via a parallel-axial dual-spinneret have been moved to Fig. 3b.

Reviewer: - *referencing is reasonable but some basic review articles of responsive polymers should be added (e.g., Nat. Mater., 2010,9, 101-113)*

Response: In the revised manuscript, we have added more references in the introduction of the background. As suggested by the reviewer, we have read the review paper on the emerging applications of stimuli-responsive polymer materials, and cited it as reference 9.

“Owing to the excellent softness, biocompatibility, stimulus diversity, and good compatibility with electrical and ionic components⁹⁻¹⁴, hydrogels are the ideal matrix for bio-integrated smart systems.” (P2, line 2-4)

“9. Stuart, M. A. C. et al. Emerging applications of stimuli-responsive polymer materials. *Nat. Mater.* **9**, 101-113 (2010).” (P19)

Reviewers' Comments:

Reviewer #2:

Remarks to the Author:

Much improved, can be published after minor revisions:

- Figure SI43 is good but confusing: the gap interrupts data flow but not explained in caption (I guess night break?)
- I suggest to use "bad solvent" instead of "antisolvent", it is more common in polymer literature, through the text and figures
- STD are added and rounding is done mostly fine. However, few glitches are still here. One example: it is meaningless to provide averaging for strain like 2627% +- 124%. You have to round to significant digits considering STD. Correct representation, e.g., should be 2630% +- 120%. Pls look at any statistical data analysis text and make proper rounding in few tables and text.

Dear Editor,

We thank you for your reply on the decision of our manuscript “Macromolecule conformational shaping for extreme mechanical programming of polymorphic hydrogel fibers” (NCOMMS-21-51572A) submitted to Nature Communications.

We are grateful for the reviewer’s comments for minor revision. According to the reviewer’s comments, the manuscript has been revised carefully. The modifications have been marked red in the revised manuscript.

We are thankful for the editorial comments, and have carefully addressed the points listed in the Author Checklist to comply with the editorial requests.

Your consideration of our manuscript for publication is highly appreciated and we look forward to hearing from you.

Yours sincerely,
Ghim Wei Ho, Cantab, FRSC
Professor, Vice dean of Engineering
National University of Singapore
E-mail: elehw@nus.edu.sg

REVIEWER COMMENTS

Reviewer #2 (Remarks to the Author):

Much improved, can be published after minor revisions:

- Figure SI43 is good but confusing: the gap interrupts data flow but not explained in caption (I guess night break?)

- I suggest to use "bad solvent" instead of "antisolvent", it is more common in polymer literature, through the text and figures

- STD are added and rounding is done mostly fine. However, few glitches are still here. One example: it is meaningless to provide averaging for strain like 2627% +- 124%. You have to round to significant digits considering STD. Correct representation, e.g., should be 2630% +- 120%. Pls look at any statistical data analysis text and make proper rounding in few tables and text.

Response to reviewer comments for manuscript:

“Macromolecule conformational shaping for extreme mechanical programming of polymorphic hydrogel fibers” (NCOMMS-21-51572A)

Reviewer #2 (Remarks to the Author):

- Figure SI43 is good but confusing: the gap interrupts data flow but not explained in caption (I guess night break?)

Response: The generated V_{oc} and I_{sc} were separately monitored for over 5 hours, and the gap interrupt in the figure was caused by changing the connection of electrodes for measurement of I_{sc} . In the revised manuscript, we have explained the gap interrupt in Supplementary Fig. 43.

Supplementary Figure 43. V_{oc} and I_{sc} generated by the 500% strain Janus spring on the hot tube filled with circulating water of $\sim 90^\circ\text{C}$. The generated V_{oc} and I_{sc} were separately monitored for over 5 hours, and the gap interrupt in the figure was caused by changing the connection of electrodes for measurement of I_{sc} .

- I suggest to use "bad solvent" instead of "antisolvent", it is more common in polymer literature, through the text and figures.

Response: We thanks for the review's suggestion. We have reviewed the literature, and "antisolvent precipitation", "antisolvent crystallization" and "antisolvent approach" are commonly used. We prefer to keep our method as "antisolvent phase separation".

- STD are added and rounding is done mostly fine. However, few glitches are still here. One example: it is meaningless to provide averaging for strain like 2627% +- 124%. You have to round to significant digits considering STD. Correct representation, e.g., should be 2630% +- 120%. Pls look at any statistical data analysis text and make proper rounding in few tables and text.

Response: As suggested by the reviewer, we have make rounding to the calculated data with decreased precision. The changes in the revised manuscript now reads:

“The pH-dependent mechanical parameters (Supplementary Fig. 16 and Table S1), correspond to the scope of elongation ratio of 105% ± 2% to 2630% ± 120%, tensile strength of 1210 ± 120 kPa to 47 ± 5 MPa, modulus of 240 ± 30 kPa to 2050 ± 370 MPa, and toughness of 1.7 ± 1.1 MJ m⁻³ to 17.8 ± 1.6 MJ m⁻³.”

Table S1. Mechanical properties of the hydrogel microfibers of different pH. Data are presented as average value with standard deviation.

	Breaking strain (%)	Breaking stress (MPa)	Elastic modulus (MPa)	Toughness (MJ m ⁻³)
pH 3.95	5 ± 2	47 ± 5	2050 ± 370	1.7 ± 1.1
pH 5.12	390 ± 60	6.7 ± 0.5	32.8 ± 3.6	15.4 ± 3.3
pH 6.35	850 ± 40	1.26 ± 0.07	0.28 ± 0.02	3.6 ± 0.1
pH 9.14	1090 ± 50	1.25 ± 0.07	0.24 ± 0.03	4.8 ± 0.3
pH 12.38	1540 ± 120	1.22 ± 0.04	0.27 ± 0.03	9.7 ± 1.5
pH 13.34	2530 ± 120	1.21 ± 0.12	0.34 ± 0.03	17.8 ± 1.6
pH 13.97	590 ± 80	2.5 ± 0.2	23.3 ± 6.5	11.9 ± 1.8